# 2 + 1 dimensional Floquet systems and lattice fermions: Exact bulk spectral equivalence

Thomas Iadecola[1,2]★, Srimoyee Sen[1]† and Lars Sivertsen[1]‡

**1** Department of Physics and Astronomy, Iowa State University, Ames, Iowa 50011, USA
**2** Ames National Laboratory, Ames, Iowa 50011, USA

★ iadecola@iastate.edu , † srimoyee08@gmail.com , ‡ lars@iastate.edu

## Abstract

A connection has recently been proposed between periodically driven systems known as Floquet insulators in continuous time and static fermion theories in discrete time. This connection has been established in a (1 + 1)-dimensional free theory, where an explicit mapping between the spectra of a Floquet insulator and a discrete-time Dirac fermion theory has been formulated. Here we investigate the potential of static discrete-time theories to capture Floquet physics in higher dimensions, where so-called anomalous Floquet topological insulators can emerge that feature chiral edge states despite having bulk bands with zero Chern number. Starting from a particular model of an anomalous Floquet system, we provide an example of a static discrete-time theory whose bulk spectrum is an exact analytic match for the Floquet spectrum. The spectra with open boundary conditions in a particular strip geometry also match up to finite-size corrections. However, the models differ in several important respects. The discrete-time theory is spatially anisotropic, so that the spectra do not agree for all lattice terminations, e.g. other strip geometries or on half spaces. This difference can be attributed to the fact that the static discrete-time model is quasi-one-dimensional in nature and therefore has a different bulk-boundary correspondence than the Floquet model.

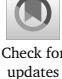 Check for updates

# 1 Introduction

Periodically driven quantum systems known as Floquet insulators have emerged as a new paradigm for nonequilibrium phases and phase transitions [1–7]. Despite their non-equilibrium nature, their stroboscopic dynamics—in which the system is observed at integer multiples of the drive period $T$—defines the notion of a quasi-energy spectrum that facilitates their analysis. They can manifest special phases of matter that have no analogs in equilibrium systems. The simplest example of this can be found in fermionic topological insulators [8]. At equilibrium, these systems are known to host zero energy edge states in $1+1$ dimensions or gapless states on the boundary in higher dimensions [9–11]. These localized edge modes are in one-to-one correspondence with the bulk of the insulator being in a nontrivial topological phase. If the bulk undergoes a transition to a non-topological phase, the boundary modes disappear. In the case of a $1+1$ dimensional Floquet insulator, one can engineer additional topological phases besides the ones observed in static systems. When the bulk of the material is in one of these out-of-equilibrium topological phases, the boundary can host modes which cannot be classified as zero energy modes any longer. For example, there can be modes with quasi-energy $\pi/T$ called $\pi$ modes [12–16], whose existence is protected by spectral symmetries unique to the Floquet setting. In particular, since the quasi-energy spectrum is defined by taking the logarithm of the time evolution operator at the drive time, it is invariant under shifts by integer multiples of the drive frequency $2\pi/T$. Thus, $\pi/T$ is special since it is invariant (modulo $2\pi/T$) under the transformation $\epsilon \to -\epsilon$ where $\epsilon$ stands for the quasi-energy.

This conventional notion distinguishing Floquet systems and static systems, while accurate in continuous time, is incomplete as it does not account for static systems in discrete time. The latter is commonplace in lattice field theory where quantum field theories are typically formulated on a spacetime lattice. In these systems, time discretization leads to the appearance of new states analogous to $\pi$ modes in Floquet systems. The simplest example of this can be found by considering a Dirac fermion in discrete spacetime. The equation of motion (EOM), i.e. the Dirac equation, has the form of a Schrödinger equation due to appearance of only a linear time derivative. The equation can be rewritten in the form of a discrete-time Schrödinger equation,

$$\sum_{t'} i \nabla_{t,t'} \psi_{t'} = H \psi_t \,, \tag{1}$$

where $H$ is a static Dirac Hamiltonian, $\nabla_{t,t'} = \frac{\delta_{t,t'-\tau} - \delta_{t,t'+\tau}}{2\tau}$ is the symmetric finite difference operator on a temporal lattice with lattice spacing $\tau$ and $\hbar = 1$. When $H$ is topological (which occurs in an appropriate parameter regime), there exist zero energy edge states, with wavefunction $\phi$, such that $H\phi = 0$. It is now easy to see that Eq. (1) is satisfied by $\phi$ as well as $\phi e^{i(\frac{\pi}{\tau})t}$. The latter is the analog of the Floquet $\pi$ mode since it corresponds to a frequency of $\pi/\tau$. The appearance of these $\pi$ modes in lattice field theory goes by the name of fermion doubling [17–22].

Given the similarities between the two systems, it is natural to explore whether they share some exact mathematical equivalence. As shown in Refs. [23, 24] such an equivalence indeed exists in $1+1$ dimensions, where we demonstrated that the Floquet spectrum of a certain driven system with period $T$ can be reproduced by a static Dirac fermion theory in discrete time with lattice spacing $\tau = T$. In detail the $1+1$ dimensional correspondence found in Refs. [23, 24] goes as follows. The Floquet Schrödinger equation is given by $i\partial_t \chi = H_F \chi$ where $H_F$ is the Floquet Hamiltonian extracted from the time evolution operator evaluated at time $T$:

$$H_F = i \frac{\ln U_F(T)}{T} \,. \tag{2}$$

Fourier transforming the Floquet Schrödinger equation to frequency space, one finds stationary

solutions with frequency $k_0 = \epsilon$, where $\epsilon$ are the eigenvalues of the Floquet Hamiltonian, i.e., the quasienergies. Refs. [23, 24] demonstrated that there exists a static lattice fermion action defined in discrete time with lattice spacing $\tau = T$ which can reproduce the quasi-energy spectrum of the Floquet system. The discrete time lattice action leads to a classical EOM of the form

$$D\psi = 0,\tag{3}$$

where $D$ is some differential operator composed of spatial and temporal finite difference operators. One can Fourier transform in time to write $D$ in frequency space. The zero eigenvalues of the operator $D$ correspond to a set of frequency values $k_0$ which match the Floquet quasi-energies $\epsilon$. Note that in discrete-time theories the frequency variable is periodic with a periodicity of $2\pi/\tau$. Thus the quasi-energy periodicity of the Floquet spectrum matches the periodicity of the frequency variable in the discrete time setup when the time lattice spacing $\tau$ is equated with the drive period $T$.

In this paper, we explore the question of equivalence between Floquet systems and static topological Hamiltonians in 2+1 dimensions. Our ultimate goal is to identify the criteria under which such equivalences can be found between the Floquet systems and lattice fermions more generally. However, such a discussion is beyond the scope of this paper. The goal of this paper instead, is to explore whether such equivalences can be found at all in 2+1 dimensions. Therefore, we explore one of the simplest nontrivial Floquet model in 2+1 dimensions studied in [25]. Our hope is that studies like these will help uncover the deeper principles underlying the equivalence between the Floquet systems and discrete time setups.

The topology of Floquet systems in 2+1 dimensions becomes much richer than in 1+1 dimensions, including the existence of so-called anomalous Floquet topological insulators [25]. These systems can exhibit topologically protected gapless boundary modes even when bulk bands are nontopological according to the classification scheme for static systems. We consider an anomalous Floquet system in two spatial dimensions that can exhibit chiral edge states and come up with a static discrete time lattice action that reproduces several important features of the Floquet system, including the locations of its topological phase transitions.

At the same time, there are important differences between the Floquet and discrete-time lattice theory constructed in this paper. First, the static Hamiltonian is anisotropic in contrast with the Floquet Hamiltonian. Surprisingly, this anisotropy does not show up in the eigenvalues of the static Hamiltonian with periodic boundary conditions (PBC), which allows the PBC Floquet spectrum to be reproduced exactly in the static system. However, the anisotropy affects the spectrum with open boundary conditions (OBC). Specifically, when considering a strip geometry that is periodic in one direction and open in the other, the spectrum of the static discrete-time Hamiltonian only matches that of the Floquet system when the strip is oriented in a particular direction. This of course is unsurprising given the anisotropic nature of the Hamiltonian. However, the match between spectra with PBC and in the appropriate strip geometry is analytically exact. This exact analytic match under the two geometries is the main result of this paper.

Second, the Floquet system is chiral in the sense that, when the system is defined in a strip geometry, each edge carries edge states of a specific chirality such that two opposite edges together produce a Dirac fermion. In contrast, the static discrete-time Hamiltonian in the strip geometry has nonchiral massless Dirac edge states on each edge. The nonchiral edge states of the discrete-time theory are protected by a static topological invariant, while the chiral edge states of the anomalous Floquet system are protected by an intrinsically Floquet topological invariant.

This observation poses an interesting question. Chiral edge states occur quite naturally in discrete space-time lattice fermions. However, as described in the main text of the paper, the spectra of these discrete space-time theories don't match that of the Floquet system discussed

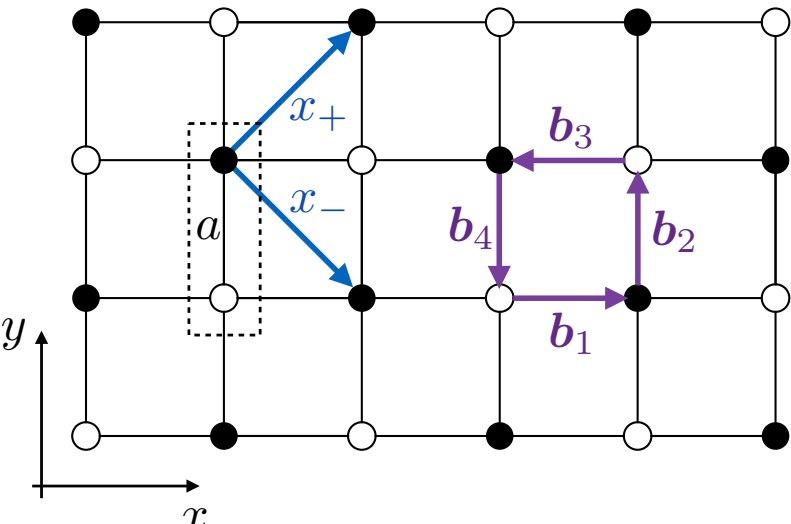

Figure 1: Sketch of the lattice on which the Floquet model is defined. A and B sublattice sites are colored in black and white, respectively. The dashed box indicates the unit cell of the lattice, and the blue vectors indicate the directions of the lattice vectors of the corresponding Bravais lattice. The purple nearest-neighbor vectors $\boldsymbol{b}_i$, $i = 1, \ldots, 4$ indicate the directions of the hoppings during each of the four time steps within a Floquet cycle.

in this paper, one to one. A natural follow up question is if the spectra can be made to agree in the infrared and whether the bulk boundary correspondence of the Floquet system and that of the discrete space-time static theory can be related in a rigorous way. This too is beyond the scope of this paper and will be explored in future work.

The organization of the paper is as follows. We begin in Sec. 2 with a description of the Floquet system including the driving protocol and the quasi-energy spectrum with PBC and OBC. We then construct in Sec. 3 a static Hamiltonian which replicates some of the essential features of the Floquet spectrum. In the process, we demonstrate an exact analytic correspondence between the PBC and OBC spectra of the two systems. We conclude in Sec. 4 with a summary and an outlook for future work.

## 2 The Floquet model

We begin with a square lattice shown in Fig. 1 with two sublattices A and B colored in black and white, respectively. We will use a driving protocol defined in Ref. [25] in which fermions perform discrete cyclotron-like motion driven by an alternating pattern of hoppings around the plaquettes of the square lattice (see Fig. 1).

In momentum space, the time-dependent Hamiltonian for this driving is given by

$$H(t) = \sum_{\mathbf{k}} \begin{pmatrix} c_{\mathbf{k},A}^\dagger & c_{\mathbf{k},B}^\dagger \end{pmatrix} H(\mathbf{k}, t) \begin{pmatrix} c_{\mathbf{k},A} \\ c_{\mathbf{k},B} \end{pmatrix}, \tag{4}$$

$$H(\mathbf{k}, t) \equiv \sum_{n=1}^{4} H_i = \sum_{n=1}^{4} J_n(t) \Big[ e^{i\mathbf{b}_n \cdot \mathbf{k}} \sigma_+ + e^{-i\mathbf{b}_n \cdot \mathbf{k}} \sigma_- \Big], \tag{5}$$

$$\sigma_+ = \frac{1}{2}(\sigma_x + i\sigma_y), \qquad \sigma_- = \frac{1}{2}(\sigma_x - i\sigma_y), \tag{6}$$

where $c^\dagger_{\mathbf{k},l}$ and $c_{\mathbf{k},l}$ are creation and annihilation operators acting on sublattice $l \in \{A, B\}$ and $\sigma_{x,y,z}$ are the usual 2 by 2 Pauli matrices. Here, the vectors $\mathbf{b}_n = (a, 0), (0, a), (-a, 0), (0, -a)$ specify the direction of the hopping at time steps $n = 1, \dots, 4$ within a Floquet cycle, where $a$ is the lattice spacing. We continue by defining the hopping amplitude

$$J_n(t) = \begin{cases} J, & \text{if } \frac{(n-1)T}{4} \le t < \frac{nT}{4}, \\ 0, & \text{otherwise,} \end{cases} \qquad 1 \le n \le 4, \tag{7}$$

where $T$ is the total period of the driving.

Now that we have defined the driving Hamiltonian, we can write down the time evolution operator at time $T$, also known as the Floquet operator:

$$U(T) = U_4 U_3 U_2 U_1 = e^{-iH_4 T/4} e^{-iH_3 T/4} e^{-iH_2 T/4} e^{-iH_1 T/4}, \tag{8}$$

$$H_1 = J \left( e^{iak_x} \sigma_+ + e^{-iak_x} \sigma_- \right),$$
$$H_2 = J \left( e^{iak_y} \sigma_+ + e^{-iak_y} \sigma_- \right),$$
$$H_3 = J \left( e^{-iak_x} \sigma_+ + e^{iak_x} \sigma_- \right),$$
$$H_4 = J \left( e^{-iak_y} \sigma_+ + e^{iak_y} \sigma_- \right). \tag{9}$$

Since $U_n$ is a sum of Pauli matrices, we can write the evolution during each time step in the following way:

$$U_n = \cos(JT/4) - i \sin(JT/4) \left[ \cos(\mathbf{b}_n \cdot \mathbf{k}) \sigma_x - \sin(\mathbf{b}_n \cdot \mathbf{k}) \sigma_y \right]. \tag{10}$$

It is now straightforward to obtain the $U(T)$, the Floquet Hamiltonian and the quasienergies using Eq. (2),

$$\cos(\epsilon T) = \frac{1}{4} \Big[ 3 + \cos JT - (1 - \cos JT) \cos(ak_x - ak_y) - (1 - \cos JT) \cos(ak_x + ak_y)$$
$$- (1 - \cos JT) \cos(ak_x - ak_y) \cos(ak_x + ak_y) \Big], \tag{11}$$

which are shown in Fig. 2 for $JT = 1.5\pi$. Note that the quasi-energy spectrum is gapless at $\epsilon = 0$, but gapped around $\epsilon = \pi/T \pmod{2\pi/T}$. As shown in Ref. [25], this system exhibits phase transitions at drive periods $T$ that are odd multiples of $\pi/J$. The first Brillouin zone extends from $-\pi/(\sqrt{2}a) < k_\pm < \pi/(\sqrt{2}a)$ where $k_\pm \equiv (k_x \pm k_y)$.

To study the OBC spectrum of this model, and to expedite later comparisons to lattice fermion theories, it is convenient to rewrite the position-space Hamiltonian in terms of the Bravais lattice coordinates $x_\pm$ (see Fig. 1). The corresponding Floquet eigenvalues can be obtained from (11) by substituting $(k_x \pm k_y) \to \sqrt{2}k_\pm$. This focuses our attention on the first Brillouin zone of the model, which is highlighted with a contour plot in Fig. 3.

Under this change of variables, and fixing the lattice spacing via $2a^2 = 1$, the quasi-energy spectrum under PBC becomes

$$\cos(\epsilon T) = \frac{1}{4} \Big[ 3 + \cos JT - (1 - \cos JT) \cos k_+$$
$$- (1 - \cos JT) \cos k_- - (1 - \cos JT) \cos k_+ \cos k_- \Big]. \tag{12}$$

To obtain the OBC spectrum, we write down the position space Hamiltonian in one of the directions $i = \pm$. The OBC spectrum of this system (with open boundary in $x_-$) as a function of $k_+$ is given in the Figs. 4 and 5, where we used 6 transverse lattice sites in the $x_-$ direction and set the driving period to $T = 1.5\pi/J$ and $0.5\pi/J$, respectively. In the case of $JT = 1.5\pi$, we find one chiral/Weyl edge mode on each boundary. Edge modes that are located on the

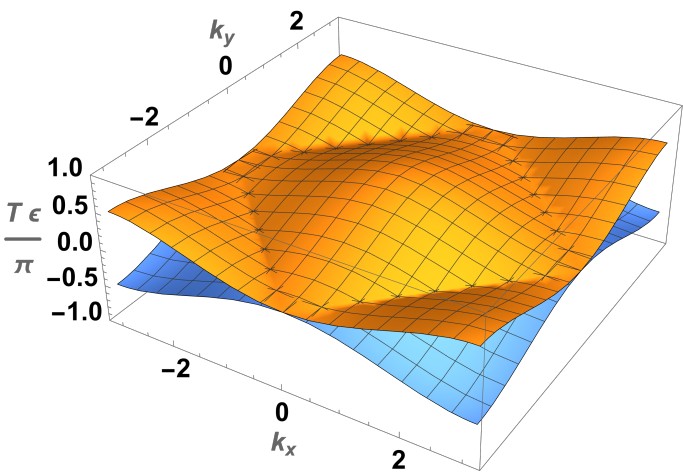

Figure 2: PBC Floquet eigenvalues plotted as a function of momentum $k_{x,y}$ in units of $1/a$ for parameter value $JT = 1.5\pi$.

same boundary are colored in the same color, either red or green; edge modes of opposite chiralities live on opposite edges.

In the case, $JT = 0.5\pi$, the spectrum is very similar, except that there are no edge modes, indicating that the parameter values $JT = 1.5\pi$ and $JT = 0.5\pi$, are separated by a bulk phase transition.

# 3 Construction of the static Hamiltonian

At this point we can ask the question: *Can this Floquet spectrum be replicated by the spectrum of a discrete-time model?*. In other words, if the Floquet Schrödinger equation is expressed as

$$i\partial_t \psi = H_F \psi, \tag{13}$$

where $H_F$ is the Floquet Hamiltonian with quasi-energy eigenvalues $\epsilon$, we ask whether the quasi-energy eigenvalues can be reproduced using a discrete time Schrödinger equation with a static Hamiltonian, $H_s$ and time lattice spacing $T$, e.g.

$$\sum_{t'} i\nabla_{t,t'}\phi(x,t') = H_s\phi(x,t), \tag{14}$$

where $\nabla_{t,t'} = \frac{\delta_{t,t'-T} - \delta_{t,t'+T}}{2T}$ is the naively discretized symmetric finite difference operator. We will take the spatial lattice spacing (and in some cases the unit cell spacing) in the target static theory to be 1. Let us denote the eigenvalues of $H_s$ as $\epsilon_s$. Fourier transforming to frequency-momentum space, Eq. (14) becomes

$$\frac{1}{T}\sin(k_0 T)\phi(k_0,k) = H_s\phi(k_0,k). \tag{15}$$

For a finite-size lattice, the implementation of a Fourier transform assumes PBC. Fixing $|\phi\rangle$ to be an eigenstate of $H_s$ with eigenvalue $\epsilon_s$, we find that Eq. (15) has two solutions:

$$\begin{aligned}
k_0 &= \frac{1}{T}\sin^{-1}(\epsilon_s T), \\
k_0 &= \frac{\pi}{T} - \frac{1}{T}\sin^{-1}(\epsilon_s T).
\end{aligned} \tag{16}$$

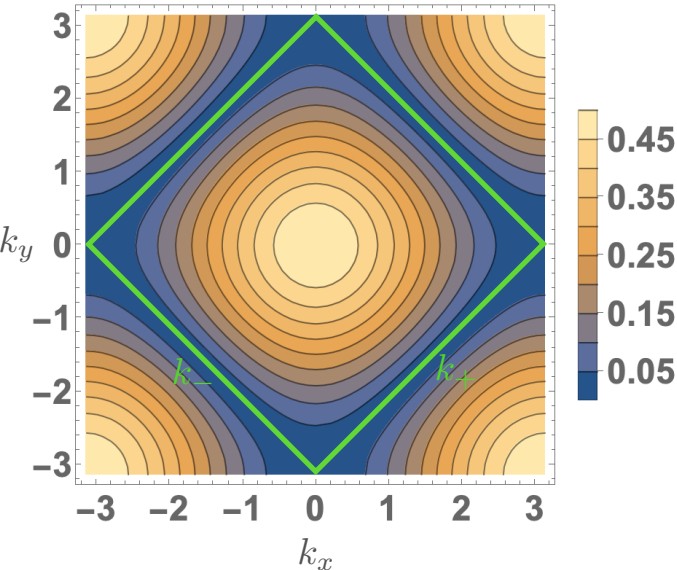

Figure 3: Contour plot of the quasi-energy eigenvalues as a function of momenta $k_x, k_y$ in units of $1/a$ at $JT = 1.5\pi$. The Brillouin zone, enclosed by the gapless contour highlighted in green, is indexed using momenta $-\pi/(\sqrt{2}a) < k_\pm < \pi/(\sqrt{2}a)$ where $k_\pm \equiv (k_x \pm k_y)$.

This doubling of the number of solutions to Eq. (15) relative to the static Schrödinger equation, known as fermion doubling, is a consequence of the naive replacement of the continuous-time derivative with the discrete-time one in Eq. (14). Demanding that the allowed values of $k_0$ are in one-to-one correspondence with the quasi-energy eigenvalues $\epsilon$ seems to require that the quasi-energy spectrum exhibit $\pi$-pairing. That is, for any quasi-energy value $\epsilon$, there must be a $\pi$-partner (or, in the fermion-doubling language, a doubler mode) at $\pi/T - \epsilon$. This feature is not generically present in the quasi-energy spectra of Floquet systems, and indeed is not a feature of Eq. (12). Making a more generic connection between Floquet systems and discrete-time theories therefore requires us to discretize time in a way that avoids fermion doubling.

One way to achieve this is to broaden our motivating question to: *can the Floquet quasi-energy eigenvalues be reproduced from the classical equations of motion derived from the action of some discrete-time theory? or can the Floquet spectra be used to construct a discrete time static lattice fermion theory whose phase transitions and edge spectra match that of the Floquet system?* For instance, the naively discretized Schrödinger equation (14) can be traced back to an action of the form

$$S = \int_{x,t} \phi^\dagger(i\partial_t - H_s)\phi \,, \tag{17}$$

where the classical EOM is in fact the Schrödinger equation. Discretization of space and time followed by replacement of the time derivative with the symmetric finite difference operator results in Eq. (14). The frequency-momentum space form of the above action is

$$S = \int_{k_0,k} \phi^\dagger\left(\frac{1}{T}\sin(k_0 T) - H_s(\vec{k})\right)\phi \,, \tag{18}$$

where $H_s(\vec{k})$ is the momentum space form of $H_s$. This is formally referred to as naive time discretization in lattice field theory. The fermion doubling/ $\pi$-pairing in this theory is a result of the naive time discretization. To break this pairing, one could introduce higher dimensional temporal derivative operators in the lattice action, e.g. one that introduces a $\cos k_0$ dependence in frequency space, reminiscent of the Wilson term in Euclidean lattice field theory. [23, 24].

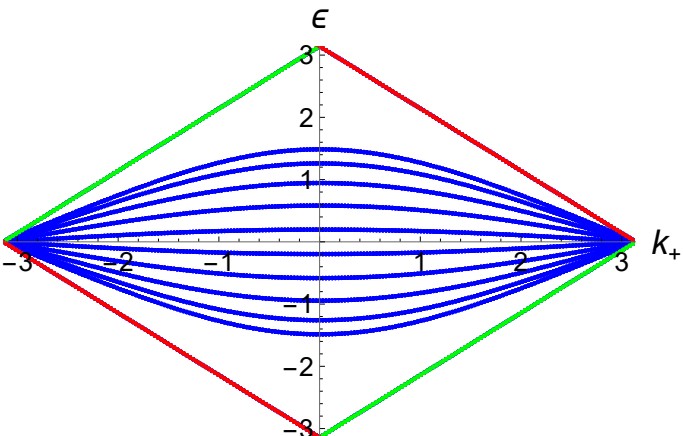

Figure 4: OBC Floquet eigenvalues plotted in units of $1/T$ at $JT = 1.5\pi$. Chiral edge modes crossing the gap at $\epsilon = \pm\pi/T$ are colored in red and green; modes of the same color live on the same spatial boundary.

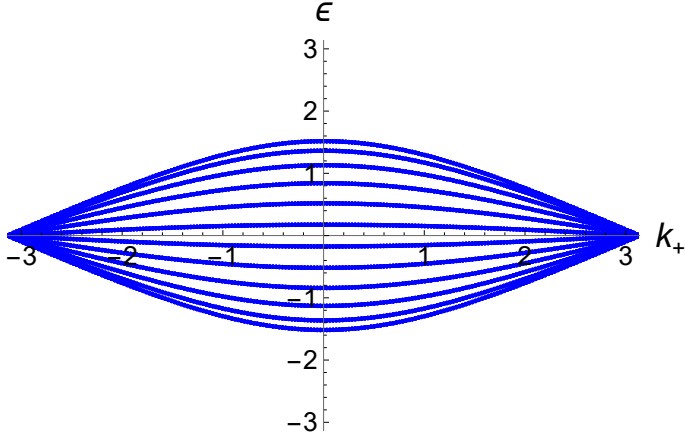

Figure 5: OBC Floquet eigenvalues plotted in units of $1/T$ at $JT = 0.5\pi$. In this case there are no edge modes.

In this paper we work with a *staggered* discrete-time derivative to break the $\pi$ pairing. This is defined on a temporal lattice with a two-site unit cell. $\phi$ then has to be promoted to a two-component object, $\phi \to (\phi_+, \phi_-)$, with $\phi_+$ living on one sublattice and $\phi_-$ living on the other. The distance between the two sites in a unit cell is $T/2$. $\phi_\pm$ can themselves be composed of multiple components commensurate with the dimension of $H_s$. Since the two components live on different time sites, the corresponding action is sometimes referred to as staggered [26]. The frequency space form of this derivative is given by

$$d(k_0) \equiv \frac{1}{2T}\begin{pmatrix} 0 & 1 - e^{ik_0 T} \\ 1 - e^{-ik_0 T} & 0 \end{pmatrix}, \tag{19}$$

where this two dimensional matrix acts on the two components $\phi_\pm$. This form of the derivative can be expressed in the time lattice using an operator of the form

$$d = (-1)^i \frac{\left(\delta_{i,j-1} - \delta_{i,j+1}\right)}{2}, \tag{20}$$

where $i, j$ are now denoting time lattice sites. It's simple to derive this result by implementing a unit cell inverse Fourier transform in the time direction. We will not give the explicit derivation

here. However, we will discuss the derivation of an analogous spatial derivative in subsection 3.2. It is now clear that the operator $d$ is completely local in time. To see that this is a legitimate way of describing a time derivative, one can diagonalize $d(k_0)$ and take the small $k_0$ limit at which point the eigenvalues have the form $\pm k_0/2$ which in the continuum can be expressed as $\pm i\partial_t/2$. The corresponding action can be written as

$$S = \int_{k_0,\vec{k}} \phi^\dagger \left(1 \times d(k_0) - H_s \otimes 1_{2\times 2}\right) \phi\,, \tag{21}$$

where the $2 \times 2$ identity matrix in the second term is explicitly keeping track of the two-site unit cell and the identity operator appearing on the first term of the above equation has the same dimensions as $H_s$. This results in the EOM

$$\frac{1}{T}\sin(k_0 T/2)(1 \otimes \sigma_3)\varphi = (H_s \otimes 1_{2\times 2})\varphi\,, \tag{22}$$

where $\varphi = S\phi$ such that $S(d(k_0))S^\dagger = \frac{1}{T}\left(\sin\frac{k_0 T}{2}\right)\sigma_3$. The solutions in this case have the form

$$k_0 = \frac{2}{T}\sin^{-1}(\epsilon_s T)\,, \tag{23}$$

and there are (generically) no $\pi$ paired solutions. However, the two site unit cell introduces additional degeneracy in the solutions as indicated by the explicitly written tensor product of $H_s$ with $1_{2\times 2}$ in Eq. (22). To reproduce the Floquet quasi-energy eigenvalues with solutions to Eq. (23), one can simply set $\epsilon_s = \frac{1}{T}\sin(T\epsilon/2)$.

Our next task is to construct the Hamiltonian $H_s$ for which this correspondence holds. Naively, this task seems easy—e.g., one could substitute $H_s$ with the following Hamiltonian $H$:

$$H(k) = \frac{1}{T}\begin{pmatrix} \sin(T\epsilon(k)/2) & 0 \\ 0 & -\sin(T\epsilon(k)/2) \end{pmatrix}. \tag{24}$$

Here $\epsilon(k)$ stands for the quasi-energy eigenvalues of the Floquet quasi-energy spectrum under PBC. Eq. (23) then trivially reproduces the PBC Floquet spectrum. However, we demand more from our target Hamiltonian: we require that the target Hamiltonian constructed using PBC Floquet eigenvalues should also reproduce the quasi-energy eigenvalues under OBC. For the latter we will need an expression for the Hamiltonian in position space. Clearly, the Hamiltonian in Eq. (24) does not contain much information about the correct position space form. One could naively attempt to obtain a position space form of the Hamiltonian in Eq. (24) by taking an inverse Fourier transform of each of the diagonal entries of $H$. However, $\frac{1}{T}\sin(T\epsilon(k)/2)$ will in general include complicated functions of momenta including square roots which when Fourier transformed may yield a very non-local matrix in position space.

In fact, since the Floquet OBC spectrum may contain topologically protected edge states, to replicate the same kind of edge state behavior our target Hamiltonian must also be topological. However, a static topological Hamiltonian under PBC is expected to be gapped whereas the Floquet quasi-energy spectrum considered here, under PBC, is gapless (see Fig. 2). This implies that $\frac{1}{T}\sin(\epsilon(k)T/2)$ would include zero eigenvalues. Therefore, it appears impossible to reproduce the Floquet spectrum using a static topological Hamiltonian.

At this point we take advantage of the fact that the Floquet spectrum is periodic in quasi-energy with periodicity $2\pi/T$, and that the spectrum has a gap around quasi-energy $\pi/T$. The periodicity in quasi-energy allows us to relabel the spectrum such that it becomes amenable to being reproduced using a static Hamiltonian. In the next subsection we will give a schematic discussion of how this relabeling works. It will also include an overview of the final results describing the correspondence between the Floquet quasi-energy spectrum and discrete time spectrum.

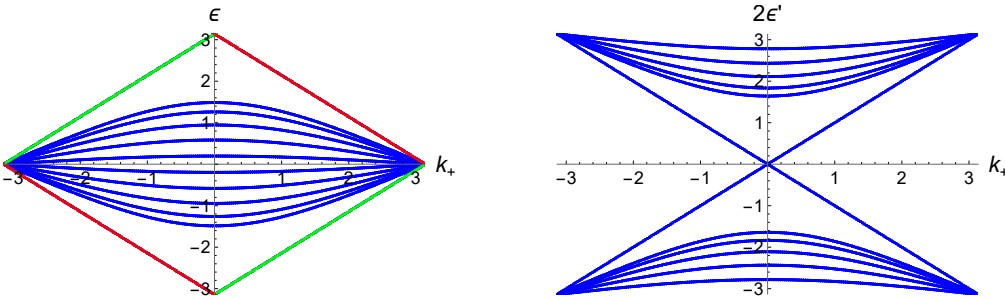

Figure 6: On the left we have a plot of the Floquet quasi-energy eigenvalues plotted in units of $1/T$ for $JT = 1.5\pi$ with OBC using 6 site lattice in $x_-$ direction. (This is a reproduction of Fig. 4 to facilitate comparison with the right-hand panel.) On the right we plot the $\pi$-shifted quasi-energies $2\epsilon'_\pm$ as described in the text in units of $1/T$.

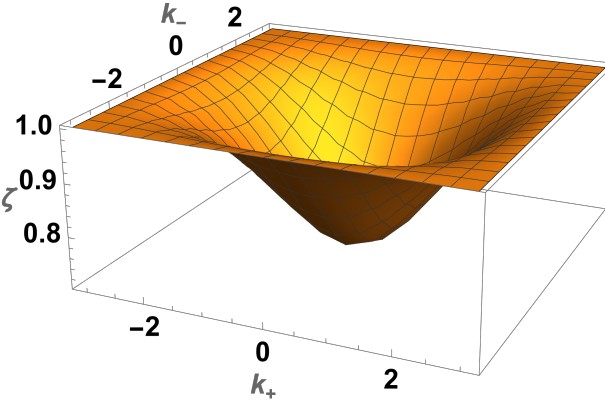

Figure 7: PBC eigenvalues of the Hamiltonian, $\zeta$, defined in Eq. (27), in units of $1/T$ for a representative $JT = 1.5\pi$.

## 3.1 Overview

Here, we outline the correspondence between the Floquet quasi-energy eigenvalues and the discrete-time spectrum of our target Hamiltonian without giving the details of the Hamiltonian itself, which will be provided in the next subsection. We use our final results for the target spectrum to explain the correspondence and to guide the eye of the reader. We consider the OBC spectrum here for expository purposes, but the mapping we develop also holds exactly for PBC.

In Fig. 6 we show the Floquet quasi-energy eigenvalues for $JT = 1.5\pi$ (left) and the solutions to the discrete-time EOM (23) (right) with OBC. Here, by OBC we mean a strip geometry that is periodic in the $x_+$ direction and open in the $x_-$ direction; the plots in Fig. 6 use 6 sites in the $x_-$ direction. The spectrum shown in the right subfigure is an exact match for that in the left panel, except that the former is shifted by $\pi/T$ compared to the latter. This relabeling/shifting of the spectrum is needed to engineer a gap around zero frequency ($k_0 = 0$) for the discrete-time theory. The frequency solutions in the right panel and quasienergies in the left panel both have a periodicity of $2\pi/T$. This is the essence of the correspondence, i.e., the Floquet spectrum is reproduced exactly using a discrete time theory if we identify quasi-energy zero with frequency $k_0 = -\pi/T$ in the discrete time theory and quasi-energy $\pi/T$ with $k_0 = 0$ in the discrete time theory.

As mentioned earlier, in this paper we will work with the action of Eq. (21) and the frequency solutions (23). Therefore, the $\pi$-shifted quasienergies shown in the right subfigure of

Fig. 6 have to equal $\frac{2}{T}\sin^{-1}(\epsilon_s T)$. In order to construct our target Hamiltonian, we have to invert the resulting equation to obtain $\epsilon_s$, the eigenvalues of the target static Hamiltonian. To illustrate this in detail we define

$$
\begin{aligned}
\epsilon'_+ &= \frac{\epsilon_+}{2} - \frac{\pi}{2}, \\
\epsilon'_- &= \frac{\epsilon_-}{2} + \frac{\pi}{2}.
\end{aligned}
\tag{25}
$$

where $\epsilon_\pm$ are the positive and negative branches of the original quasi-energy spectrum (with PBC or OBC) and $2\epsilon'_\pm$ are the $\pi$-shifted positive and negative quai-energy eigenvalues. At times we will drop the $\pm$ in the subscript to refer to an eigenvalue without specifying its sign. The $\pi$-shifts appearing in the definition of $2\epsilon'$ perform the reassignment of quasi-energy $\pi/T$ with 0 and 0 with $-\pi/T$ mentioned above. $\epsilon_s$ is then given by

$$
\epsilon_s = \frac{1}{T}\sin(\epsilon' T).
\tag{26}
$$

## 3.2 Details of the mapping

We now describe the process of constructing $H_s$ which will allow us to arrive at the correspondence outlined in the previous subsection. The first step is writing down the PBC eigenvalues of the target Hamiltonian from Eq. (26)

$$
\zeta \equiv \pm \frac{1}{T}\sqrt{\frac{3+m^2}{4} - \frac{1-m^2}{4}\cos k_+ - \frac{1-m^2}{4}\cos k_- - \frac{1-m^2}{4}\cos k_+ \cos k_-},
\tag{27}
$$

where $m = \cos(JT/2)$. Here we could have used the symbol $\epsilon_s$ instead of $\zeta$. However, we allow $\epsilon_s$ to represent the eigenvalues of the target Hamiltonian under both PBC and OBC depending on the context. Therefore, it is better to assign a different symbol to refer specifically to the PBC eigenvalues. Note that, in every figure where we show the spectra of the target lattice Hamiltonian, except in Fig. 7, we will be illustrating the results with open boundary condition (OBC). Hence the axes labeling will indicate $\epsilon_s$ or $\sin(T\epsilon')$ in all figures except Fig. 7.

We show only the positive values of $\zeta$ in the Fig. 7 for a representative parameter value $T = 1.5\pi$. This spectrum has a gap and the eigenvalues have a chance of being mapped to a static topological Hamiltonian. Doing the same for the OBC eigenvalues, we obtain Fig. 8 (plotted for $T = 1.5\pi$). Therefore, our goal is to construct a static Hamiltonian whose PBC eigenvalues match those of Fig. 7, (or equivalently Eq. (27)) and whose OBC eigenvalues in a strip geometry match those in Fig. 8. If this is accomplished, then we will have a static Hamiltonian whose discrete-time spectrum reproduces the Floquet spectrum under both periodic and open boundary conditions, i.e. Figs. 2, 4, and 5, albeit shifted vertically by $\pi/T$ (as shown in the right subfigure of Fig. 6).

It is important to note that $T\zeta$ reaches 1 for any $k_\pm = -\pi, \pi$. This feature will significantly constrain the class of static Hamiltonians we can write down. Additionally, the edge spectrum with linearly dispersing edge states will put further constraints on the target Hamiltonian.

To see how these constraints can be nontrivial to satisfy, consider a $2+1$ dimensional Wilson-Dirac Hamiltonian which in lattice field theory is known to exhibit chiral edge states on its boundary. The Hamiltonian is given by

$$
H_{\text{WD}} = R(i\gamma_+ \nabla_+ + i\gamma_- \nabla_-) + \gamma_0\left(M - \frac{R}{2}\Delta\right).
\tag{28}
$$

Here, $\nabla_\pm$ are symmetric finite difference operators in the spatial directions $x_\pm$ conjugate to $k_\pm$: $\nabla_\pm = \frac{\delta_{x_\pm, x'_\pm - 1} - \delta_{x_\pm, x'_\pm + 1}}{2}$. $\Delta$ is the symmetric finite difference Laplacian $\Delta = \Delta_+ + \Delta_-$ given by $\Delta_\pm = \delta_{x_\pm, x'_\pm - 1} + \delta_{x_\pm, x'_\pm + 1} - 2\delta_{x_\pm, x'_\pm}$. $\gamma_\pm$ are anticommuting, Hermitian matrices satisfying

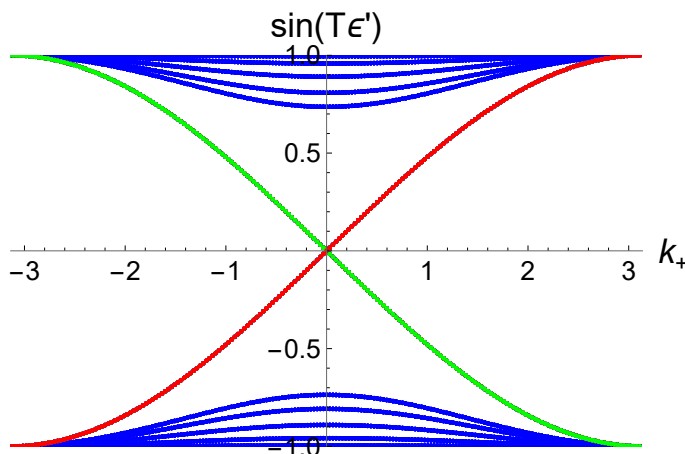

Figure 8: Eigenvalues of the desired target static Hamiltonian with OBC in $x_-$ for $JT = 1.5\pi$. These are given by Eq. (26) where we use OBC Floquet eigenvalues for $\epsilon_\pm$. We again use $N_- = 6$ transverse sites to make this plot.

$\{\gamma_i, \gamma_j\} = 2\delta_{ij}$ where $i, j$ can be $\pm, 0$. In momentum space, the eigenvalues of this Hamiltonian have the form

$$e_{\text{WD}} = \pm\sqrt{R^2 \sum_{j=\pm}\left(\sin^2 k_j\right) + \left(M + R\left(\sum_{j=\pm}(1 - \cos k_j)\right)\right)^2}. \qquad (29)$$

If we set $k_+ = \pm\pi$, we obtain

$$e_{\text{WD}}(k_+ = \pm\pi) = \pm\sqrt{M^2 + 6MR + 10R^2 - 2R(M + 3R)\cos k_-}. \qquad (30)$$

Clearly, this expression includes $k_-$ dependence in contrast with the eigenvalues described by Eq. (27) which equal $1/T$ for any $k_-$ when $k_+ = \pm\pi$. Therefore, to make the eigenvalues $e_{\text{WD}}$ consistent with Eq. 27, we will have to set $M = -3R$ as well as $M^2 + 6MR + 10R^2 = 1/T$. Finally, to be consistent with the eigenvalue at $k_+ = k_- = 0$, one has to set $M = \pm m/T$. All three of these conditions cannot be satisfied simultaneously which rules out the Wilson-Dirac Hamiltonian as the target Hamiltonian.

To further explore whether the PBC and OBC eigenvalues displayed in Figs. 7 and 8 can be reproduced using a static fermionic topological Hamiltonian, it is expedient to consider the OBC spectrum since it contains information about the edge states as well as the bulk spectrum which in turn contains information about the form of the Hamiltonian. Therefore, our goal will be to construct a Hamiltonian which reproduces the OBC edge spectrum with open boundary in one of the two directions, e.g. $x_-$. To construct this Hamiltonian, we will relax that the condition of isotropy, i.e. we will not demand that the Hamiltonian be invariant under exchanging $x_+ \longleftrightarrow x_-$ or $k_+ \longleftrightarrow k_-$.

The procedure we follow is this:

1. Propose a general form of the Hamiltonian under periodic boundary condition in both $x_-$ and $x_+$ which allows us to write this general form in terms of the momenta $k_-$, $k_+$.

2. Consider open boundary in $x_-$ while retaining PBC in $x_+$ and demand that the edge state dispersion with respect to $k_+$ reproduces the Floquet edge state dispersion of Fig.8. As we will see, this will only fix the part of the Hamiltonian which contains the edge state dispersion.

3. Then, demand that the full Hamiltonian under PBC (which allows us to express $H_s$ as a function of $k_\pm$ ) reproduce the eigenvalues of of Fig. 7.

4. Next we explore whether there exists a topological Hamiltonian which is consistent with the previous two steps, i.e. it has the edge state spectrum demanded by step 2 and it has the PBC eigenvalue behavior demanded by step 3. This is because, without a topological Hamiltonian, it does not make sense to discuss the possibility of edge states. This step is implemented retaining PBC in $x_+$ and introducing PBC in $x_-$.

5. Let's assume that such a Hamiltonian is found at which point one should be able to construct a map between the Floquet parameters and the parameters of the static Hamiltonian. However, there is still a remaining ambiguity. We note that, the parameters of such a static Hamiltonian can be dialed to undergo a phase transition and reach a part of the parameter space where the Hamiltonian is trivial/non-topological and yet has the same PBC spectra as the original Hamiltonian. Thus, PBC spectra alone will not reveal the correct map between the Floquet system and the static lattice Hamiltonian. However, the OBC spectra with OBC in $x_-$ must be different between the two non-topological and topological parts of the parameter space and we can leverage this to obtain the map between the Floquet spectra and the static Hamiltonian.

6. The previous steps are all performed while taking PBC in $x_+$ and writing the Hamiltonian in terms of $k_+$. While these steps produce a certain $k_+$ space form of the Hamiltonian, this may not be local in $x_+$. The final step is to ensure locality of the Hamiltonian in $x_+$ while maintaining the same the PBC and OBC (in $x_-$) eigenvalues.

**step 1:** We implement step 1 first by parameterizing the target Hamiltonian under PBC as follows:

$$H_s = \sum_i \gamma_i F_i \,, \tag{31}$$

where $i$ takes integer values $i = 1, 2, \dots n$ where $n$ is an arbitrary integer to be fixed later, and where $\gamma_i, F_i$ are matrices of arbitrary dimensions. They can have lattice site indices as well as indices corresponding to some internal space. We impose $[\gamma_i, F_j] = 0$ for all $i, j$ and $\{\gamma_i, \gamma_j\} = 2\delta_{ij}$, under PBC. $\gamma_i$ and $F_i$ by themselves are not necessarily local. However, we demand that $\gamma_i F_i$ be so. This allows us to open the boundaries of $H_s$. Therefore, every time we mention open boundary spectra of $H_s$ we will be opening the boundary of products like $\gamma_i F_i$ and not of $\gamma_i$ and $F_i$ individually.

We posit that $F_i$ are matrices of the form $1_{\alpha \times \alpha} \otimes f_i$ where $f_i$ are $N^2 \times N^2$ matrices that have lattice-site indices ($N$ being the number of lattice sites or unit cells depending on the context). The identity matrix in the tensor product can correspond to some internal space index or a sublattice index. Further assuming that $f_i$ are diagonal in momentum space with PBC, the bulk eigenvalue spectrum of the Hamiltonian is $\pm\sqrt{\sum_i f_i^2}$.

**Step 2**: At this point we are agnostic about the exact form of the $\gamma_i$ and $f_i$. To fix these, we need to focus on the sine of the primed quasi-energy eigenvalues of Fig. 8. One of the interesting features in the OBC spectrum of primed eigenvalues is that for $m < 0$ there are gapless edge states with a linear spectrum $k_0 = \pm\frac{k_+}{2}$ or $T\epsilon_s = \pm\sin(k_+/2)$. To replicate this behavior using our static Hamiltonian we consider $N_-$ sites along $x_-$ with open boundary. (We will take $N_- = 6$ for making figures.) The eigenstates of our static Hamiltonian are now labeled as $|\chi(k_+, l)\rangle$ where $l$ replaces the momentum label $k_-$ since we have broken translational invariance in $x_-$ by opening the boundary. Therefore $l$ can be thought of as flavor/species index taking values from 1 to $N_-$. To reproduce the linearly dispersing edge state of the Floquet spectrum, we can demand one of these species eigenstates of the target

theory, e.g. the state $|\chi(k_+, l = 1)\rangle$, satisfies

$$H_s\big|_{\text{OBC in } x_-} |\chi(k_+, 1)\rangle = \left(\sum_{i=1}^{n} \gamma_i F_i\right)\bigg|_{\text{OBC in } x_-} |\chi(k_+, 1)\rangle = \pm\frac{\sin(k_+/2)}{T}|\chi(k_+, 1)\rangle, \qquad (32)$$

where the choice of $\pm$ has no effect on the bulk spectrum. A simple way to engineer this is to enforce

$$\left(\sum_{i=2}^{n} \gamma_i F_i\right)\bigg|_{\text{OBC in } x_-} |\chi(k_+, 1)\rangle = 0, \qquad (33)$$

which forces $|\chi(k_+, 1)\rangle$ to be a zero mode of $\sum_{i=2}^{n} \gamma_i F_i$ under OBC in $x_-$. In our construction we already have $\gamma_1$ anti-commuting with $\sum_{i=2}^{n} \gamma_i F_i$ under PBC. We now demand the same under OBC in $x_-$. As a result, we can take $|\chi(k_+, 1)\rangle$ to be an eigenstate of $\gamma_1$. Therefore we introduce the states $|\chi_\pm(k_+, 1)\rangle$ where the subscript denotes whether the state has $\gamma_1$ eigenvalue $+1$ or $-1$. At this point we can set

$$f_1 = \frac{\sin(k_+/2)}{T}, \qquad (34)$$

which implies that the states $|\chi_\pm(k_+, 1)\rangle$ are linearly dispersing modes with $|\chi_+\rangle$ being a positive chirality state (defined as having a $+1$ eigenvalue of $\gamma_1$) and $|\chi_-\rangle$ being a negative chirality state.

It is already clear from this discussion that the target Hamiltonian cannot be isotropic under exchange of $k_+ \leftrightarrow k_-$ (or $x_+ \leftrightarrow x_-$). This is simply because, in order for the Hamiltonian to be isotropic, for some $i \neq 1$ we must have $f_i = \pm\frac{\sin k_-/2}{T}$ when using PBC for both directions, implying that the magnitude of the PBC Hamiltonian eigenvalues is lower-bounded by $\frac{1}{T}\sqrt{\sin^2\frac{k_+}{2} + \sin^2\frac{k_-}{2}}$. However, this lower bound exceeds $1/T$ for $k_+ = \pi$ for any $k_-$ (and vice versa). Therefore the lower bound exceeds the maximum reached by the PBC eigenvalues shown in Eq. (27). This enforces $f_{i\neq1} \neq \pm T^{-1}\sin k_-/2$ implying that the Hamiltonian is anisotropic under $k_+ \leftrightarrow k_-$.

Note that we have forced $|\chi_\pm(k_+, 1)\rangle$ be a zero mode of the operator $\sum_{i=2}^{n} \gamma_i F_i = H_s - \gamma_1 F_1$ under OBC in $x_-$ irrespective of the value of $k_+$. However, we also want $|\chi_\pm\rangle$ to be localized on the boundary in $x_-$ for all $k_+$. This will allow us to fix $H_s - \gamma_1 F_1$. Since $|\chi_+\rangle$ and $|\chi_-\rangle$ have opposite chiralities, if they are localized on opposite boundaries with OBC in $x_-$, the boundary theory is chiral. If they end up being localized on the same boundary, then the boundary theory is Dirac-like.

**Step 3:** Because the target Hamiltonian is anisotropic, we expect that its eigenvalues with PBC $[\epsilon_s(k_+, k_-)]$ may not be invariant under $k_+ \leftrightarrow k_-$ either, in which case $\epsilon_s(k_+, k_-) \neq \zeta(k_+, k_-)$ since $\zeta$ [Eq. (27)] is invariant under $k_+ \leftrightarrow k_-$. Despite this, we explore whether we can find an anisotropic Hamiltonian which, while reproducing the correct OBC behavior in $x_-$, has isotropic PBC eigenvalues so as to match the expression for $\zeta$ exactly. This forces us to demand $\zeta^2(k_+, k_-) = \sum_i f_i^2(k_+, k_-)$. Since we have fixed $f_1 = \frac{\sin k_+/2}{T}$, we find

$$\begin{aligned}
\sum_{i=2}^{n} f_i^2(k_+, k_-) &= \zeta^2(k_+, k_-) - \frac{\sin^2 k_+/2}{T^2} \\
&= \frac{\left(\frac{1+m^2}{4}(1 + \cos k_+) - \frac{1-m^2}{4}(1 + \cos k_+)\cos k_-\right)}{T^2} \\
&= \frac{\cos^2(k_+/2)}{T^2}\left(\frac{1+m^2}{2} - \frac{1-m^2}{2}\cos k_-\right).
\end{aligned} \qquad (35)$$

**Step 4:** Remarkably, the expression in the parentheses can be expressed as the square of the eigenvalues of a one dimensional lattice Dirac Hamiltonian, also known as a Wilson-Dirac Hamiltonian, of the form [23, 24]

$$H_D = \sigma_i R \sin k_- + \sigma_j \left[ m_0 + R(1 - \cos k_-) \right], \tag{36}$$

with

$$R = \frac{1 - m_0}{2}, \quad -\frac{1 + m_0}{2},$$
$$m_0 = \pm m, \tag{37}$$

where $\sigma_i$ and $\sigma_j$ are Pauli matrices with $i \neq j$, e.g. we pick $i = 1, j = 2$. The expression in parentheses can also be expressed as eigenvalues of a Su-Schrieffer-Heeger (SSH) Hamiltonian [27] of the form (in momentum space)

$$H_{\text{SSH}} = \begin{pmatrix} 0 & v + w e^{ik_-} \\ v + w e^{-ik_-} & 0 \end{pmatrix}, \tag{38}$$

with

$$v = \frac{(-1 - m)}{2}, \quad w = \frac{1 - m}{2},$$
$$v = \frac{(1 + m)}{2}, \quad w = \frac{-1 + m}{2},$$
$$v = \frac{(1 - m)}{2}, \quad w = \frac{-1 - m}{2},$$
$$v = \frac{(-1 + m)}{2}, \quad w = \frac{1 + m}{2}. \tag{39}$$

Both of these Hamiltonian can exhibit topological phases and can host zero energy edge states for appropriate choice of parameters. The ability to represent the lattice theory in terms of both Wilson-Dirac and SSH Hamiltonians is reminiscent of the Floquet-to-lattice mapping introduced in 1+1 dimensions [23, 24].

**Step 5:** All of the parameter assignments presented in Eqs. (37) and 39 work equally well in reproducing the PBC eigenvalues of Eq. (27). However, as we will see, all of them are not equally good when it comes to reproducing the right edge-state behavior. To understand this, note that $H_D$ and $H_{\text{SSH}}$ can both have zero modes with OBC depending on the parameters of the theory. When this occurs, the full Hamiltonian exhibits linearly dispersing edge states as shown in Eq. (32). Our goal is to choose parameters in such a way that the appearance of these linearly dispersing edge states in the static theory coincides with the same in the Floquet OBC spectrum. Only a subset of the choices presented in Eq. (37), (39) will satisfy this.

To see how this can come about, we can write $H_D$ and $H_{\text{SSH}}$ in position space by doing an inverse Fourier transform. To inverse Fourier transform the Wilson-Dirac Hamiltonian on $N_-$ lattice sites we define the unitary transformation $(P_0)_{k_-, x_-} = \frac{1}{\sqrt{N_-}} e^{(2\pi i) K_- \frac{x_-}{N_-}}$, where $K_- = 0, \ldots, N_-$ and $x_- = -\frac{N_-}{2}, \ldots, \frac{N_-}{2} - 1$ with $k_- = \frac{2\pi K_-}{N_-}$. The position space Hamiltonian $P_0^\dagger H_D P_0$ is then given by

$$H_D = i\sigma_1 R \nabla_- + \sigma_2 \left( m_0 - \frac{R}{2} \Delta_- \right), \tag{40}$$

where we have defined $\nabla_-$ and $\Delta_-$ previously after Eq. (28). To write the corresponding position space Hamiltonian in $x_-$ for the SSH Hamiltonian, we construct a $2N_- \times 2N_-$ lattice in $x_-$ which is divided into 2 sublattices or $N_-$ unit cells where each unit cell consists of 2 sites hosting a single one component fermion. We now implement a Fourier transform over

the $N_-$ unit cells by defining $F_{K_-,X_-} = \frac{1}{\sqrt{N_-}}e^{i\frac{2\pi K_-}{N_-}X_-}$ where $K_- = -\frac{N_-}{2}, -\frac{N_-}{2}+1, \ldots, \frac{N_-}{2}-1$ where $X_-$ denote unit cell coordinates and $k_- = \frac{2\pi K_-}{N_-}$. The corresponding Fourier transform matrix acting on the individual lattice sites $x_-$ is given by $(F_0)_{K_-,X_-} = \frac{1}{\sqrt{N_-}}e^{i\frac{2\pi K_-}{N_-}X_-} \otimes 1_{2\times2}$. The position space Hamiltonian $F_0 H_{\text{SSH}} F_0^\dagger$ is then given by

$$(H_{\text{SSH}})_{ij} = \frac{(1+(-1)^i)}{2}(w\delta_{i,j-1} + v\delta_{i,j+1}) + \frac{(1-(-1)^i)}{2}(v\delta_{i,j-1} + w\delta_{i,j+1}), \qquad (41)$$

where $i,j$ take values from 1 to $2N_-$. The 1D Wilson-Dirac Hamiltonian hosts zero-energy localized edge states with OBC in $x_-$ when $\frac{m_0}{R} < 0$, whereas there are no edge states for $\frac{m_0}{R} > 0$. In the former case, we can expect to find linearly dispersing edge states for the full Hamiltonian. Therefore, if we pick $m_0 = m$, then we must pick $R = \frac{1-m_0}{2}$ so as to match onto the Floquet edge-state spectrum which has edge states for $m < 0$ and no edge states for $m > 0$. Similarly, if we pick $m_0 = -m$, then we must pick $R = -\frac{1+m_0}{2}$ so as to find edge states for $m < 0$ and no edge states for $m > 0$. Either of these two choices work. We pick $m_0 = m$ and $R = \frac{1-m_0}{2}$ and call the corresponding Wilson Dirac Hamiltonian $H_A$. Similarly, the SSH Hamiltonian hosts zero-energy edge states for $|w| > |v|$. There are two choices for which this condition maps onto the nontrivial Floquet edge spectrum, or $m < 0$:

$$v = \frac{(-1-m)}{2}, \qquad w = \frac{1-m}{2},$$
$$v = \frac{(1+m)}{2}, \qquad w = \frac{-1+m}{2}. \qquad (42)$$

We pick the latter and call the corresponding SSH Hamiltonian $H_B$. Given the observations of the previous paragraphs, the simplest choice for $H_s - \gamma_1 F_1$ appears to be $\frac{\cos(k_+/2)H_A}{T}$ or $\frac{\cos(k_+/2)H_B}{T}$, with $\gamma_1 = \sigma_k$ and $k \neq i,j$ of Eq. (36).

**Step 6:** However, the appearance of $\sin(k_+/2)$ and $\cos(k_+/2)$ in the Hamiltonian introduces non-locality in $x_+$. To remedy this, one can introduce two sublattices in the $x_+$ direction and stagger the fermion Hamiltonian the following way. We define

$$h_1 \equiv \sin(k_+/2)\begin{pmatrix} 0 & -ie^{ik_+/2} \\ ie^{-ik_+/2} & 0 \end{pmatrix} = \frac{1}{2}\begin{pmatrix} 0 & 1-e^{ik_+} \\ 1-e^{-ik_+} & 0 \end{pmatrix}, \qquad (43)$$

$$h_2 \equiv \cos(k_+/2)\begin{pmatrix} 0 & e^{ik_+/2} \\ e^{-ik_+/2} & 0 \end{pmatrix} = \frac{1}{2}\begin{pmatrix} 0 & 1+e^{ik_+} \\ 1+e^{-ik_+} & 0 \end{pmatrix}, \qquad (44)$$

where we have doubled the fermionic degrees of freedom exactly as we did for the time lattice in the context of $d(k_0)$ in Eq. (19). The $2 \times 2$ space of the matrices $h_1, h_2$ corresponds to the two lattice sites of the unit cell.

With this, we propose for the full Hamiltonian

$$H_{s,A} = \frac{h_1 \otimes 1 + h_2 \otimes H_A}{T}, \qquad (45)$$

or

$$H_{s,B} = \frac{h_1 \otimes 1 + h_2 \otimes H_B}{T}, \qquad (46)$$

where in the tensor product of $h_1$ with the identity matrix: 1 has the dimensions of $H_A$ or $H_B$ depending on the context.

In other words, we set

$$\gamma_1 = \begin{pmatrix} 0 & -ie^{ik_+/2} \\ ie^{-ik_+/2} & 0 \end{pmatrix} \otimes 1_{2\times 2},$$

$$f_1 = \frac{\sin k_+/2}{T},$$

$$F_1 = \frac{\sin k_+/2}{T} \otimes 1_{2\times 2} \otimes 1_{2\times 2} = \begin{pmatrix} \frac{\sin k_+/2}{T} & 0 \\ 0 & \frac{\sin k_+/2}{T} \end{pmatrix} \otimes 1_{2\times 2}. \tag{47}$$

The last line in the equation for $F_1$ is meant to clarify how the matrix product of $\gamma_1 F_1$ is meant to be taken. It is easy to read off the other $\gamma_i$, $f_i$ and $F_i$. E.g. for $H_{s,A}$ we can write

$$\gamma_2 = \begin{pmatrix} 0 & e^{ik_+/2} \\ e^{-ik_+/2} & 0 \end{pmatrix} \otimes \sigma_1,$$

$$f_2 = \left(\frac{1-m}{2T}\right)\cos(k_+/2)\sin(k_-),$$

$$F_2 = \left(\frac{1-m}{2T}\right)(\cos(k_+/2) \otimes 1_{2\times 2}) \otimes (\sin(k_-) \otimes 1_{2\times 2})$$

$$= \left(\frac{1-m}{2T}\right)\begin{pmatrix} \cos(k_+/2) & 0 \\ 0 & \cos(k_+/2) \end{pmatrix} \otimes \begin{pmatrix} \sin(k_-) & 0 \\ 0 & \sin(k_-) \end{pmatrix},$$

$$\gamma_3 = \begin{pmatrix} 0 & e^{ik_+/2} \\ e^{-ik_+/2} & 0 \end{pmatrix} \otimes \sigma_2,$$

$$f_3 = \frac{m}{T} + \frac{(1-m)}{2T}(1 - \cos k_-),$$

$$F_3 = \begin{pmatrix} f_3 & 0 \\ 0 & f_3 \end{pmatrix} \otimes 1_{2\times 2}, \tag{48}$$

and all other $\gamma_i F_i = 0$ for $i > 3$. Again, the last equality on $F_2$ and $F_3$ are meant to clarify how the products $\gamma_2 F_2$ and $\gamma_3 F_3$ need to be taken. Similarly for $H_{s,B}$,

$$\gamma_2 = \begin{pmatrix} 0 & e^{ik_+/2} \\ e^{-ik_+/2} & 0 \end{pmatrix} \otimes \sigma_1,$$

$$f_2 = \left(\frac{m\cos^2 k_-/2}{T} + \frac{\sin^2 k_-/2}{T}\right)\cos(k_+/2),$$

$$F_2 = \begin{pmatrix} \cos(k_+/2) & 0 \\ 0 & \cos(k_+/2) \end{pmatrix} \otimes \begin{pmatrix} \frac{m\cos^2 k_-/2}{T} + \frac{\sin^2 k_-/2}{T} & 0 \\ 0 & \frac{m\cos^2 k_-/2}{T} + \frac{\sin^2 k_-/2}{T} \end{pmatrix},$$

$$\gamma_3 = \begin{pmatrix} 0 & e^{ik_+/2} \\ e^{-ik_+/2} & 0 \end{pmatrix} \otimes \sigma_2,$$

$$f_3 = \frac{m-1}{2T}\cos(k_+/2)\sin k_-,$$

$$F_3 = \frac{m-1}{T}\begin{pmatrix} \cos(k_+/2) & 0 \\ 0 & \cos(k_+/2) \end{pmatrix} \otimes \begin{pmatrix} \cos(k_-/2) & 0 \\ 0 & \cos(k_-/2) \end{pmatrix}. \tag{49}$$

Just as before, the expressions for $F_2$ and $F_3$ clarify how the products $\gamma_2 F_2$ and $\gamma_3 F_3$ are to be taken. The advantage of this choice lies in the fact that $h_1$ and $h_2$ can be implemented using a staggered-fermion Hamiltonian or the SSH model, ensuring locality in $x_+$. This will become evident as we write out the position space form of $h_1$ and $h_2$ next. Interestingly, the form of $h_1$ and $h_2$ has an impact on the linearly dispersing Dirac edge states that we found with OBC

in $x_-$. $h_1$ and $h_2$ by themselves correspond (say) to the SSH Hamiltonian at the massless point, e.g. $h_1 = H_{\text{SSH}}\big|_{w=1,v=-1}$ and $h_2 = H_{\text{SSH}}\big|_{w=1,v=1}$. Therefore, the linearly dispersing boundary Dirac fermions correspond to the massless SSH model $h_1$. These massless edge Dirac fermions are protected by the chiral symmetry (i.e., sublattice symmetry) of $h_1$.

For completeness, we write the position space Hamiltonian in $x_+$, by following the procedure outlined in obtaining Eq. (41). We construct a lattice of $2N_+$ sites in the $x_+$ direction, divided into $N_+$ unit cells with each unit cell consisting of 2 sites. We now implement an inverse discrete Fourier transform over the $N_+$ unit cells by defining $F'_{K_+,X_+} = \frac{1}{\sqrt{N_+}} e^{i\frac{2\pi K_+}{N_+}X_+}$ where $K_+ = -\frac{N_+}{2}, -\frac{N_+}{2}+1, \ldots, \frac{N_+}{2}-1$ with $X_+$ denoting the unit cell positions and $k_+ = \frac{2\pi K_+}{N_-}$. The Fourier transform matrix acting on the individual lattice sites $x_+$ is given by $(F'_0)_{K_+,X_+} = \frac{1}{\sqrt{N_+}} e^{i\frac{2\pi K_+}{N_+}X_+} \otimes 1_{2\times 2}$. The Fourier transforms $F'_0 h_1 (F'_0)^\dagger$ and $F'_0 h_2 (F'_0)^\dagger$ yield the position-space forms of $h_1$ and $h_2$:

$$(h_1)_{ij} = (-1)^i \frac{\left(\delta_{i,j-1} - \delta_{i,j+1}\right)}{2}, \tag{50}$$

$$(h_2)_{ij} = \frac{\left(\delta_{i,j-1} + \delta_{i,j+1}\right)}{2}, \tag{51}$$

where $i, j$ label lattice sites. Both are local in $x_+$. This completes all the six steps outlined. So, we have succeeded in constructing two anisotropic, local Hamiltonians, Eqs. (45), (46), with the same PBC eigenvalues that are completely isotropic.

**The complete structure of $H_{s,A}$, $H_{s,B}$:** We review the complete structure of $H_{s,A}$ and $H_{s,B}$ for the readers' convenience. The position space form of $H_{s,A}$ on the spatial lattice is given by

$$\left(H_{s,A}\right)_{ik,jl} = \frac{(h_1)_{ij} \otimes 1_{kl} + (h_2)_{ij} \otimes (H_A)_{kl}}{T}, \tag{52}$$

where the subscript $k, l$ correspond to the spatial lattice index in the $x_-$ direction as well as an associated "spin"/internal space index as shown in Eq. 53 below. The subscript $i, j$ correspond to the lattice site indices in the $x_+$ direction. The position space form of $H_A$ is given by

$$(H_A)_{kl} = (H_D)_{kl}\big|_{m_0=m, R=\frac{1-m}{2}} = \left(i\frac{1-m}{2}\sigma_1 \nabla_- + \sigma_2\left(m - \frac{1-m}{2}\Delta_-\right)\right)_{kl}, \tag{53}$$

and there is an implied tensor product between the Pauli matrices and the operators $\Delta_-, \nabla_-$ defined on the $x_-$ spatial lattice. Also, there is an implicit identity matrix of the same dimension as $\nabla_-$ and $\Delta_-$ multiplied with the parameter $m$ inside the inner parenthesis. The identity operator in Eq. 52 in tensor product with $h_1$ has the same dimension as $H_A$. $h_1$ and $h_2$ are given by Eq. 50 and 51. Similarly, the position space expression for $H_{s,B}$ is given by

$$\left(H_{s,B}\right)_{ik,jl} = \frac{(h_1)_{ij} \otimes 1_{kl} + (h_2)_{ij} \otimes (H_B)_{kl}}{T}, \tag{54}$$

with

$$(H_B)_{kl} = \frac{(1+(-1)^k)}{2}(w\delta_{k,l-1} + v\delta_{k,l+1}) + \frac{(1-(-1)^k)}{2}(v\delta_{k,l-1} + w\delta_{k,l+1}), \tag{55}$$

where $k, l$ correspond to the spatial lattice index in the $x_-$ and take values from 1 to $2N_-$. The parameter values for $v$ and $w$ are given by $v = \frac{1+m}{2}$ and $w = \frac{1-m}{2}$. As before, the subscript $i, j$ correspond to the lattice site indices in the $x_+$ direction.

The Hamiltonians of Eq. (45) and (46) were constructed to reproduce the linearly dispersing edge state of the Floquet spectrum with OBC in $x_-$. However, this construction in fact

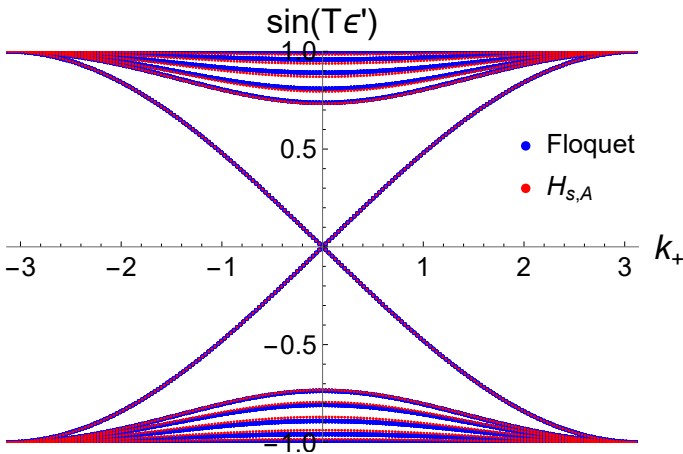

Figure 9: Eigenvalues of Eqs. (45) and (46) with OBC in the $x_-$ direction (red) for parameters $m_0, R_0, w, v$ corresponding to $JT = 1.5\pi$. The number of sites in $x_-$ is $N_- = 6$. The blue lines compare this data with the spectrum from Fig. 8.

reproduces the full spectrum with OBC in $x_-$. In Fig. 9, we plot the eigenvalues of Eq. (45) and 46 with OBC in the $x_-$ direction in red. The blue lines compare this data with the spectrum from Fig. 8. We find an exact match up to $\sim 1/N_-$ corrections, indicating that the spectra match exactly in the thermodynamic limit $N_- \to \infty$. We find that each edge hosts a single massless Dirac fermion for $m < 0$. For $m > 0$ we find no edge states. This implies that the bulk undergoes a phase transition as $m$ passes through zero. This transition occurs exactly where $(JT = \pi)$ a similar transition takes place in the Floquet system and the corresponding edge state behavior in the Floquet spectrum changes. Therefore, we see that the phase transitions of the Floquet model correspond with equivalent transitions in the static discrete-spacetime theory for the parameter choices made above.

We can now count the number of states which for a specific momenta have the same energy. We can call this number 'flavors'. The spectrum of $H_{s,A/B}$ contains two flavors with OBC and PBC. With OBC there a single massless Dirac-like fermion located on each edge. Taking the two edges together we get a two flavors of massless Dirac-like states. The two flavors results from the introduction of unit cell in $x_+$ that was necessary to maintain strict spatial locality of the discrete-time theory.

To compute the discrete time spectrum and the corresponding flavor number for this static Hamiltonian, we solve Eq. (23). The resulting solutions for $JT = 1.5\pi$ quite clearly reproduces the right panel of Fig. 6 with OBC. The PBC eigenvalues of the Floquet spectrum are reproduced the same way. If we count the the number of flavors, i.e. number of states that have the same $p_0$ solution for a specific momentum, we find that every solution is now associated with four flavors [two flavors coming from $H_s$ and another two arising from the introduction of the two sublattices in the time direction introduced to define $d(k_0)$ in Eq. (19)]. We had previously mentioned the time-lattice form of $d(k_0)$ without giving an explicit derivation of it. Given our discussion of the position space Hamiltonian, it's now easy to see by comparing Eq. (19) and Eq. (43) that $d(k_0)$ can be implemented on the time lattice using an operator analogous to Eq. (50), as shown in Eq. 20

Besides the introduction of this four flavors, the only point of divergence with the Floquet model arises from the chirality of the edge states. Whereas all edge states of one chirality in the Floquet spectrum reside on the same boundary, in the static discrete-time spectrum every wall hosts two Dirac fermions.

In summary, the linearly dispersing edge states of the Floquet system are replicated in the

discrete time theory via the edge states of the one dimensional static Hamiltonian $H_A$ or $H_B$. Every zero mode of $H_A$ or $H_B$ under OBC in $x_-$ corresponds to a branch (more precisely, two identical branches corresponding to two flavors) on the linearly dispersing edge spectrum of the full Hamiltonian $H_{s,A/B}$. Since, $H_{s,A/B}$ is anisotropic, we do not expect the edge spectrum of $H_{s,A/B}$ with OBC in $x_+$ to reproduce the linearly dispersing edge states of the Floquet model. The energy spectrum with OBC in $x_+$ is shown in Fig. 10. Interestingly, rather than chiral edge modes, we find non-dispersing zero energy edge states for all $k_-$. However, the rest of the OBC spectrum matches that of the Floquet model [Fig. 8] up to finite-size corrections $\sim 1/N_+$. It is simple to see why there are non-dispersing states in the Hamiltonian of Eqs. (45) and (46) with OBC in $x_+$. While analyzing the OBC spectrum in $x_+$, we can diagonalize $H_A$ and $H_B$ for convenience. They have eigenvalues

$$\epsilon_{A/B} = \pm\sqrt{\frac{1+m^2}{2} - \frac{1-m^2}{2}\cos k_-}\,. \tag{56}$$

With this diagonalization the Hamiltonians $H_{s,A/B}$ can be recast in the form of an SSH Hamiltonian

$$(H_{\text{SSH}})_{ij} = \frac{(1+(-1)^i)}{2}(w\delta_{i,j-1} + v\delta_{i,j+1}) + \frac{(1-(-1)^i)}{2}(v\delta_{i,j-1} + w\delta_{i,j+1})\,, \tag{57}$$

with $w = (\epsilon_{A/B} - 1)/2$ and $v = (\epsilon_{A/B} + 1)/2$. Note that the SSH model is known to have zero-energy edge states for $|w| > |v|$. Since $\epsilon_{A/B}$ takes both positive and negative values for a specific $k_-$, for every $k_-$ we have a negative $\epsilon_{A/B}$ for which the Hamiltonian will have a zero mode. This produces the non-dispersing sector in Fig. 10. Again, the spectra contain two flavors with identical dispersion just as we found in the case of Fig. 9. Moreover, in the discrete-time spectrum with staggered time derivative, we will again find a four flavors with identical discrete time spectra.

As we saw, the boundary spectrum that results from using OBC in $x_-$ reproduces the Floquet edge spectrum exactly including the bulk transition at $m = 0$. The behavior of the boundary spectrum can be connected to the bulk Hamiltonian by noticing that the transition happens when $H_A$ or $H_B$ undergoes a transition from a topologically nontrivial phase to a trivial one. $H_{A/B}$ by themselves are one dimensional Hamiltonians. Therefore the bulk transition in this case corresponds to a jump in the one-dimensional topological invariant of $H_{A,B}$. This further underscores the quasi-one-dimensional nature of $H_s$, despite its exact spectral equivalence in the bulk to the isotropic, two-dimensional Floquet model.

For the sake of completeness we also compare the OBC (in $x_-$) wave functions of the Floquet edge state and the corresponding edge states of the static Hamiltonians $H_{s,A}$ and $H_{s,B}$ for $k_+ = 0$ in Fig. 11. Using 7 lattice sites we find excellent agreement between all three.

## 4   Conclusion

This paper extends to higher dimensions previous work where we established a mathematical correspondence between one-dimensional Floquet systems and discrete-time fermion theories in one dimension. We find that some of the essential features of a certain $2+1$ dimensional anomalous Floquet spectrum can be replicated using an anisotropic static Hamiltonian in $2+1$ dimensions. This Hamiltonian can also be interpreted as a quasi one-dimensional system, i.e. even though the full static Hamiltonian is defined in $2+1$ dimensions, it is built out of two one-dimensional static Hamiltonians: $h_{1/2}$ and $H_{A/B}$. $h_{1/2}$ are defined on a lattice in the $x_+$ direction whereas $H_{A/B}$ are defined on a lattice in the $x_-$ direction.
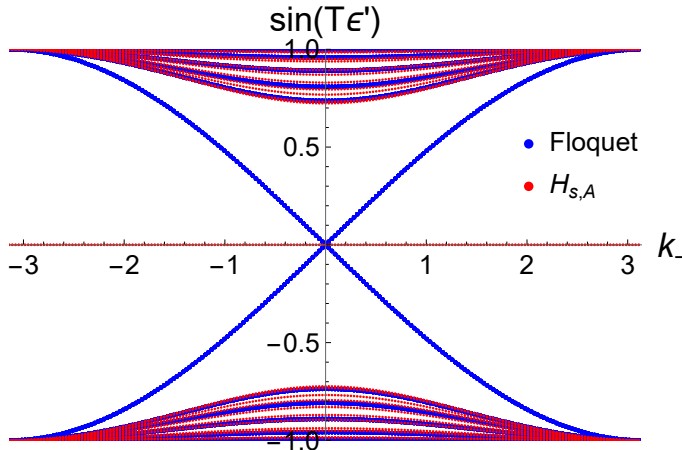

Figure 10: Eigenvalues of Eqs. (45) and (46) with OBC in the $x_+$ direction (red) for parameters $m_0, R_0, w, v$ corresponding to $JT = 1.5\pi$. The number of sites in $x_+$ is $N_+ = 6$. The blue lines compare this data with the spectrum from Fig. 8. Note that there are non-dispersing edge states along the $k_-$ axis that are not present in the Floquet model or in the strip geometry considered in Fig. 9.

The bulk-boundary correspondence pertaining to the nontrivial boundary spectrum of this static theory arises from the boundary spectrum of the one-dimensional Hamiltonian $H_{A/B}$.

The spectrum of the Floquet model with PBC matches exactly the spectrum of the discrete-time theory with PBC, modulo the appearance of four flavors in the discrete-time theory. With OBC in a strip geometry, the two spectra match when the boundary is opened in the $x_-$ direction.

Even though the bulk and boundary spectra of the Floquet and discrete-time models match, there are two essential differences between these systems. The first is that, whereas the Floquet system is completely isotropic, the static Hamiltonian is anisotropic and quasi-one-dimensional. Secondly, the Floquet boundary spectrum is chiral, which is to say that even though the strip geometry results in a massless Dirac-like edge spectrum, edge states of opposite chirality live on opposite boundaries. The static theory, while matching the massless Dirac spectrum, is not chiral. This of course is expected, given that the bulk-boundary correspondence at play is fundamentally one dimensional. Each boundary of the static system hosts an equal number of massless Dirac fermions.

These results motivate further exploration of the ties between higher-dimensional Floquet systems and discrete-time static theories. The correspondence outlined in this paper ties a gapless anomalous Floquet system to gapped static theories. It will be interesting to explore how this correspondence gets modified for gapped $2 + 1$ dimensional Floquet insulators. Furthermore, there may be other approaches to formulating a static discrete-time model that reproduces the spectrum of the Floquet model besides the one taken here. For example, while we have focused on finding a static Hamiltonian whose discrete-time spectrum is an exact match for the Floquet model, it may be possible to relax this requirement in such a way that the boundary spectrum of the Floquet model can be more faithfully reproduced in an appropriate long-wavelength limit. Going beyond $2 + 1$ dimensions, similar ties may exist between static theories and Floquet systems in higher spacetime dimensions as well and should be explored in future work. Several questions of interest remain unexplored even in one spatial dimension, e.g. whether such correspondences can be established between interacting Floquet systems and discrete-time theories and how one can relate correlators measured in Floquet systems to those measured in discrete-time lattice theories.

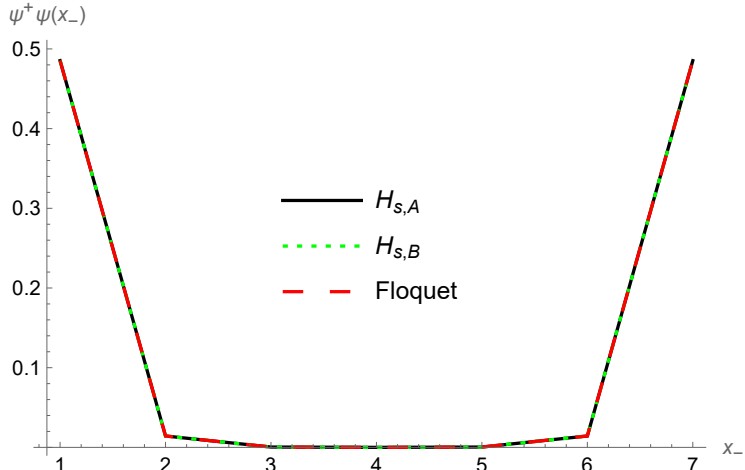

Figure 11: We plot the probability density for the wave function of the edge state at $k_+ = 0$ under OBC in $x_-$, as a function of $x_-$ for the Floquet system and the static Hamiltonians $H_{s,A}$ and $H_{s,B}$ with $JT = 1.5\pi$ using 7 lattice sites.

## Acknowledgments

**Funding information**    T.I. acknowledges support from the National Science Foundation under Grant No. DMR-2143635. S.S. and L.S. acknowledge support from the U.S. Department of Energy, Nuclear Physics Quantum Horizons program through the Early Career Award DE-SC0021892.

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
