# Peer review of "$2+1$ dimensional Floquet systems and lattice fermions: Exact bulk spectral equivalence"

_SciPost Physics Core, doi:SciPost Phys. Core 8, 035 (2025)_

## Round 1 · Referee Report · Anonymous (Referee 1) · 2024-12-18

Strengths
2-The results and its limitations are clearly stated and can be reproduced.
3-The study succeeds at extending previous research to 2D systems for a specific model.
Weaknesses
2-The information in some parts of the text lacks structure which makes it hard to read.
3-The consistency in notation, units and plot labeling can be improved.
Report
The paper is written in a mostly clear and concise way except for section 3B, which is much longer than the rest and can benefit from further structure and consistency in notation. In this section, the main thread of the explanation of the construction of the static Hamiltonian is intertwined with other considerations like the limitations of the 2D Wilson-Dirac Hamiltonian or the procedure to switch from momentum to position basis. These considerations may be relevant, but they make the reading difficult in the current structure. Also, notation and references to figures and equations is sometimes inconsistent.
The explanation is detailed and reproducible, but it is sometimes inconsistent in notation and contains typos.
Relevant previous research is correctly referenced.
The abstract and conclusion sections are clear.
The introduction section gives adequate context, but it does not provide a reason for the relevance of the specific Floquet model under consideration, which seems arbitrary. The authors get this model from reference 25, but in that publication the model has other parameters that are set to zero here. What does this specific connection between these models teach us about Floquet and static discrete time systems in general? Can this approach be generalized to all Floquet models or at least to a class of them?
Also, the results are contingent on several conditions like ignoring degeneracies and chirality of the edge modes, assuming a specific lattice termination and adding a $\pi/2$ quasienergy offset. The authors should make a stronger case defending the relevance of this specific model given these restrictions.
Requested changes
1- The words "observed" and "measured" are used in the introduction to refer to the fact that the state of a Floquet system at a discrete set of moments in time can be reproduced by a static (continuous) Hamiltonian. These words can mislead non-expert readers to believe that actual measurements are taking place at every period. The phrasing should be modified. 2- It seems that the units of the Floquet period $T$ are $\hbar$ divided by the unit of energy (which is fixed to have $J=1$). These units should be included in the text and in the labels of the figures. 3-It seems like the action approach is used to solve the fermion doubling problem in time, but all that is done in the end is to use the Susskind method to discretize the time derivative. Is the discussion about the action necessary? 4-It seems like when the staggered time lattices are introduced, the distance between $\phi_+$ and $\phi_-$ is $T/2$. This should be stated explicitly. 5-The simbols $k_0$ and $p_0$ seem to represent the same quantity in 3.3, 3.9 and other places. Why are two different symbols used? 6-The $\sin$ in the second line of Eq. 3.4 seems to be a typo. 7-In Eq. 3.14, the simbol $\zeta$ is introduced to refer to the quasienergy of the target Hamiltonian of Eq. 3.13, but in this equation, the simbol $\epsilon_s$ is used for that. Why are two different symbols used for the same thing? 8- Figures 7 and 8 both plot the eigenvalues of the target Hamiltonian for different boundary conditions. However, in the first one the vertical axis is labeled $\zeta$ and in the second one $\sin(T\epsilon')$. Both plots should be labeled the same. Also units should be provided (the units of energy seem to be fixed by $J=1$) 9- In the paragraph after Eq. 3.12, there is a typo "quai-energy". 10- The caption of Fig. 6 has typos. 11- Section 3B contains a lot of information and the aforementioned intermissions distract from the main point of the discussion. This section should be reorganised. 12- Eq. 3.16 does not fit in the width of the text. 13- Does Eq. 3.18 refer to PBC or OBC? If the answer is both, it should be explained how this is possible. 14- At the end of section 3B, the expressions of $\gamma_i$ nad $f_i$ are given, but not $F_i$. In particular it is never said how many flavors $\alpha$ the final model has. This is crucial for the reproducibility of the results. 15- The sentence "We can then demand that $\pm\sqrt{\sum_if_i^2}$ contains all the eigenvalues found in the expression for $\zeta$ in Eq. (3.14)." is confusing, there are only 2 $k$-dependent eigenvalues. Doesn't this just mean that we should demand $\zeta=\pm\sqrt{\sum_if_i^2}$? 16-The discussion in the last paragraph of the left column/first of right column in page 8 seems to imply that any $f_{i\neq1}$ different from zero would yield a lower bound of the PBC eigenvalues that surpasses $1/T$ for $k_+=\pi/a$. If this is not the case, it should be explained why. 17- In the left column of page 8 there seems to be an upper case $K_-$ that should be $k_-$. This also happens in page 9 bottom left. 18- Along section 3B, it is not clear when we are referring to OBC and when to PBC. This should be made more clear, maybe by restructuring the whole section. 19-Equation 3.28 is an equality between a matrix and an object with 1 index, this should be fixed. 20- The sentence "Opening the boundary in $x_-$ is also equivalent to introducing a domain-wall-anti-wall pair in m, as a function of $x_-$." is not clear what this means? 21- In the paragraph after 3.29 there is the expression $H_s-\gamma_1f_1$. Shouldn't this $f_1$ be $F_1$ for consistency with 3.18? In general, the way $f_i$ and $F_i$ are used in the whole text does not look consistent. 22-In Eqs. 3.35 and 3.36, $f_3$ seems to have different dimensionality than $f_1$ and $f_2$. Should they be $f_i$ or $F_i$? 23-It may be helpful to provide plots of some eigenstates (especially the edge states) for facilitating understanding. 24-In the first column of page 11 the authors write "We plot the solutions for $T = 1.5\pi$ which quite clearly reproduces the right panel of Fig. 6 with OBC." This points to a relation between Fig. 6 and some other figure which is not referred to. Do they refer to Fig. 10?
Recommendation
Ask for minor revision

---

## Round 1 · Referee Report · Anonymous (Referee 2) · 2024-12-20

Strengths
1- This paper presents a principled derivation of the static Hamiltonian matching the desired Floquet spectrum. I find the presented rationale in deriving the appropriate static Hamiltonian very thorough and likely helpful to future investigations in this direction. 2- A clear discussion of both the analogies and differences between the constructed static Hamiltonian and Floquet system are presented.
Weaknesses
1- As someone not deeply familiar with Floquet theory, the introduction did not contain a sufficiently pedagogical description of the background work in 1+1D for me to follow in a self-contained way. This work was properly cited, but as the introduction does attempt to survey this work, a little further clarity and detail would be very helpful. 2- In the derivation of the two choices of static Hamiltonians, it is quite difficult to follow the full spacetime structure of the resulting operators, due to the introduction of additional staggering partway through the derivation.
Report
Requested changes
1- Please clarify the meaning of $\epsilon$ in the first paragraph. I assume these are the quasienergies noted later on? 2- Above eq (1.2), I find the phrase "time evolution operator measured at time T" unclear. As I understood Floquet theory, the notation $U_F(T)$ should here refer to time evolution for a time interval of length T. If this is the case, please clarify this text. 3- In eq (2.2) and below, the index $i$ can easily be confused with the imaginary unit. I suggest changing this to more clear notation, e.g. the notation $n$ used in [25]. 4- In eq (3.8) and below, I don't follow what the tensor product space is precisely. I understand $H_s(p)$ as acting in the 2-dimensional space of sublattices, but what about $\sin(p_0 T/2)$ later? Please clarify this briefly here. 5- The formatting should be corrected in eq (3.16) 6- As noted above, I suggest summarizing the complete spacetime structure of the Hamiltonians in (3.32) and (3.33) after Fourier transforming back into position space, following eq (3.37) and (3.38). 7- The index structure in (3.28) is unclear, as j appears on the right side but not on the left. Likewise in (3.41). Please clarify the index structure here. 8- Waiting to describe the discrete-time structure of $d(p_0)$ does not in my view help the reader in any significant way. I suggest moving this result from eq (3.39) to near eq (3.7).
Recommendation
Ask for minor revision

---

## Round 2 · Author Response

We thank the referees for their feedback. We have made several changes to the draft as outlined in the list of changes section.

---

## Round 2 · List of Changes

We thank both referees for their suggestions on improving the text and noticing the typos.

Here are the changes we made:

In response to the specific suggestions from the first referee:

  1. Please clarify the meaning of ϵϵ in the first paragraph. I assume these are the quasienergies noted later on?

We added the meaning of \epsilon on top of the right column on page 1.

  1. Above eq (1.2), I find the phrase "time evolution operator measured at time T" unclear. As I understood Floquet theory, the notation UF(T) should here refer to time evolution for a time interval of length T. If this is the case, please clarify this text.

We have changed the word from ‘measured’ to ‘evaluated’.

  1. In eq (2.2) and below, the index ii can easily be confused with the imaginary unit. I suggest changing this to more clear notation, e.g. the notation n used in [25].

We changed the index from ‘i’ to ‘n’ as the referee suggested.

  1. In eq (3.8) and below, I don't follow what the tensor product space is precisely. I understand Hs(p) as acting in the 2-dimensional space of sublattices, but what about sin(p0T/2) later? Please clarify this briefly here.

We have clarified the tensor product. There was a typo in Eq 3.8 in the previous version. The correct version is in Eq 3.9 now.

  1. The formatting should be corrected in eq (3.16)

Corrected the formatting of eq 3.16. The new eqn is 3.17

  1. As noted above, I suggest summarizing the complete spacetime structure of the Hamiltonians in (3.32) and (3.33) after Fourier transforming back into position space, following eq (3.37) and (3.38)

We added a discussion titled “The complete structure of H_{s,A} and H_{s,b}” after Eq 3.39.

  1. The index structure in (3.28) is unclear, as j appears on the right side but not on the left. Likewise in (3.41). Please clarify the index structure here.

The referee is correct. There were typos in the index structure. We fixed it. The new eqn numbers are 3.29 and 3.45.

  1. Waiting to describe the discrete-time structure of d(p0) does not in my view help the reader in any significant way. I suggest moving this result from eq (3.39) to near eq (3.7).

We moved the discrete time structure earlier to Eq 3.7.

Here are changes we made in response to the suggestion of the second referee:

  1. The words "observed" and "measured" are used in the introduction to refer to the fact that the state of a Floquet system at a discrete set of moments in time can be reproduced by a static (continuous) Hamiltonian. These words can mislead non-expert readers to believe that actual measurements are taking place at every period. The phrasing should be modified.

The word “measured” has been changed to “evaluated” above Eq 1.2.

  1. It seems that the units of the Floquet period T are ℏ divided by the unit of energy (which is fixed to have J=1). These units should be included in the text and in the labels of the figures.

hbar has been set to 1 as in the reference 25. The convention in the literature is to write quasi-energy in units of the inverse drive period. We decided to use this convention so we could be in alignment with the literature. T does have units of 1/J. We have reinstated J everywhere in the draft instead of setting it to 1 to avoid confusion.

  1. It seems like the action approach is used to solve the fermion doubling problem in time, but all that is done in the end is to use the Susskind method to discretize the time derivative. Is the discussion about the action necessary?

The action approach is not used to solve the fermion doubling problem. In fact the action approach is the source of the fermion doubling. E.g. the replacement of the time derivative by some kind of finite difference operator is forced on us by the action approach to compute the path integral. This leads to the fermion doubling problem in the first place which then demands that to obtain a single flavor theory we would need to employ some tricks. In the Hamiltonian formulation, even when there are discrete time steps, as is the case in quantum circuits, there is no fermion doubling. This is because we are not forced to replace the time derivative with finite difference operator in this case.

  1. It seems like when the staggered time lattices are introduced, the distance between ϕ+ and ϕ− is T/2. This should be stated explicitly.

We clarified this in a sentence on top of page 5, right column, in the paragraph before Eq 3.7.

  1. The symbols k0 and p0 seem to represent the same quantity in 3.3, 3.9 and other places. Why are two different symbols used?

All the p_0 have been changed to k_0

  1. The sin in the second line of Eq. 3.4 seems to be a typo.

Referee is correct. We corrected this typo. Eq number remains the same, 3.4

  1. In Eq. 3.14, the symbol ζ is introduced to refer to the quasienergy of the target Hamiltonian of Eq. 3.13, but in this equation, the simbol ϵs is used for that. Why are two different symbols used for the same thing?

The symbol \zeta was used to specifically refer to PBC eigenvalues. \epsilon_s stands for both PBC and OBC eigenvalues of the static Hamiltonian depending on the context. We have clarified this in a discussion below Eq 3.15.

  1. Figures 7 and 8 both plot the eigenvalues of the target Hamiltonian for different boundary conditions. However, in the first one the vertical axis is labeled ζ and in the second one sin(Tϵ′). Both plots should be labeled the same. Also units should be provided (the units of energy seem to be fixed by J=1)

We have reinstated J in our formula. Also, the label is not ζ because, the plot is for open boundary eigenvalues.

  1. In the paragraph after Eq. 3.12, there is a typo "quai-energy".

We changed quasienergy to quasi-energy everywhere.

  1. The caption of Fig. 6 has typos.

Corrected typos of Fig 6.

  1. Section 3B contains a lot of information and the aforementioned intermissions distract from the main point of the discussion. This section should be reorganised.

We restructured Eq 3.b. We first outline the procedure we are following dividing it in six steps outlined in page 8. This is followed by the longer discussion where we have mentioned specifically which part of the discussion corresponds to which step.

  1. Eq. 3.16 does not fit in the width of the text.

Fixed Eq 3.16’s formatting.

  1. Does Eq. 3.18 refer to PBC or OBC? If the answer is both, it should be explained how this is possible.

Eq 3.18 has now the eqn number 3.19 and the referee is correct that the eqn makes sense under PBC. This was not clarified in the text in the previous version. We have stated explicitly that the form in Eq 3.19 is for PBC right before Eq 3.19 and also added clarification in the paragraph below Eq 3.19.

  1. At the end of section 3B, the expressions of γi and fi are given, but not Fi. In particular it is never said how many flavors α the final model has. This is crucial for the reproducibility of the results.

We have included expressions for F_i in Eq 3.35, 3.36 and 3.37. Also, we realized the word flavors does not quite express clearly what \alpha is. \alpha corresponds to the space on which the Pauli matrices act. It is better to call it internal space. We have made the clarification in The paragraph appearing just before the discussion of step 2 begins on page 9.

  1. The sentence "We can then demand that ±∑_i f_i ^2 contains all the eigenvalues found in the expression for ζ in Eq. (3.14)." is confusing, there are only 2 k-dependent eigenvalues. Doesn't this just mean that we should demand ζ=±∑_i f_i ^2 ?

We have removed this confusing sentence from the text.

  1. The discussion in the last paragraph of the left column/first of right column in page 8 seems to imply that any fi≠1 different from zero would yield a lower bound of the PBC eigenvalues that surpasses 1/T for k+=π/a. If this is not the case, it should be explained why.

This is not quite the case that for any nonzero f_(i≠1) the sum ∑f_i^2 surpasses the value of 1/T for k_+=π (assuming lattice spacing to be 1). This is because you could have a scenario f_(i≠1)∝ cos⁡〖k_+/2〗 which is the case in our mapping. In that case, the sum does not surpass 1/T.

  1. In the left column of page 8 there seems to be an upper case K− that should be k−. This also happens in page 9 bottom left.

We have fixed this issue . We still use K_+ and K_-, just before Eq 3.28, Eq 3.29 and eq 3.38. They are defined as k_±=2π K_±/N_± where N_± is the number of lattice sites or unit cells depending on the context.

  1. Along section 3B, it is not clear when we are referring to OBC and when to PBC. This should be made more clear, maybe by restructuring the whole section.

We have restructured section 3.b to explain where we are using OBC and PBC. E.g. the implementation of step 1 and step 2 at the end of page 8 and the beginning of page 9 explains where one should use PBC and where OBC. Eq 3.20 and 3.21 have subscripts that indicate that we are considering OBC in those equations. Similarly, step 3 and step 5 discussions on page 9 and 10 clarifies the use of PBC and PBC.

  1. Equation 3.28 is an equality between a matrix and an object with 1 index, this should be fixed.

Eq 3.28 from the previous version has been fixed. The new eqn number is 3.29.

  1. The sentence "Opening the boundary in x− is also equivalent to introducing a domain-wall-anti-wall pair in m, as a function of x−." is not clear what this means?

We have removed this sentence.

  1. In the paragraph after 3.29 there is the expression Hs−γ1f1. Shouldn't this f1 be F1 for consistency with 3.18? In general, the way fi and Fi are used in the whole text does not look consistent.

Yes, the referee is correct. We have changed f_1 to F_1.

  1. In Eqs. 3.35 and 3.36, f3 seems to have different dimensionality than f1 and f2. Should they be fi or Fi?

Eq 3.35 and 3.36 have been changed. We now provide the explicit expressions for F_i in them. The new eqns are numbered 3.35, 3.36, 3.37.

  1. It may be helpful to provide plots of some eigenstates (especially the edge states) for facilitating understanding.

We added a plot comparing the Floquet and lattice static Hamiltonian edge state probability density for the k_+=0 mode in Fig 11 on page 14. We also added some text just before the concusion section describing the figure.

  1. In the first column of page 11 the authors write "We plot the solutions for T=1.5π which quite clearly reproduces the right panel of Fig. 6 with OBC." This points to a relation between Fig. 6 and some other figure which is not referred to. Do they refer to Fig. 10?

We have fixed the sentence to “The corresponding solutions for JT=1.5\pi quite clearly reproduces the right panel of Fig. 6 with OBC.”

Finally, we have also added two paragraphs to the introduction section to address some of the overall comments made by the two referees.

The first additional paragraph reads:

In this paper, we explore the question of equivalence between Floquet systems and static topological Hamiltonians in 2+1 dimensions. Our ultimate goal is to identify the criteria under which such equivalences can be found between the Floquet systems and lattice fermions more generally. However, such a discussion is beyond the scope of this paper. The goal of this paper instead, is to explore whether such equivalences can be found at all in 2 + 1 dimensions. Therefore, we explore one of the simplest nontrivial Floquet model in 2 + 1 dimensions studied in [25]. Our hope is that studies like these will help uncover the deeper principles underlying the equivalence between the Floquet systems and discrete time setups.

The second reads:

This observation poses an interesting question. Chiral edge states occur quite naturally in discrete space-time lattice fermions. However, as described in the main text of the paper, the spectra of these discrete space-time theories don’t match that of the Floquet system discussed in this paper exactly, one to one. A natural follow up question is if the spectra can be made to agree in the infrared and whether the bulk boundary correspondence of the Floquet system and that of the discrete space-time static theory can be related in a rigorous way. This too is beyond the scope of this paper and will be explored in future work.

---

## Editorial Decision

published